# Nutritional Strategies Prescribed During Pregnancy and Weight Gain in Women with Gestational Diabetes Mellitus: A Systematic Review of Observational Studies

**DOI:** 10.3390/nu17010043

**Published:** 2024-12-27

**Authors:** Maria Elionês de Oliveira Araújo, Amanda Maria Lira de Lucena, Iasmim Leite Fontes, Anny Cristine de Araújo, Karla Danielly da Silva Ribeiro

**Affiliations:** 1Postgraduate Program in Applied Sciences to Women’s Health, Federal University of Rio Grande do Norte, Natal 59078-900, Brazil; maria.oliveira.112@ufrn.edu.br; 2Nutrition Graduate Program, Department of Nutrition, Federal University of Rio Grande do Norte, Natal 59078-900, Brazil; amanda.lucena.702@ufrn.br (A.M.L.d.L.); iasmimlf@gmail.com (I.L.F.); 3Postgraduate Program in Health Science, Federal University of Rio Grande do Norte, Natal 59078-900, Brazil; annycristinearaujo@gmail.com; 4Department of Nutrition, Federal University of Rio Grande do Norte, Natal 59078-900, Brazil

**Keywords:** maternal health, pregnancy, maternal weight gain, diabetes

## Abstract

**Background/Objectives**: This systematic review aims to identify diets related to weight gain in pregnant women diagnosed with gestational diabetes mellitus (GDM). **Methods**: This study was performed according to the Preferred Reporting Items for Systematic Reviews and Meta-Analyses, and its protocol was registered on the International Prospective Register of Systematic Reviews (CRD42023432322). The searches used the medical subject headings in the PubMed/MEDLINE, Web of Science, Scopus, and EMBASE databases. Studies were selected, and data were extracted by three researchers. The Newcastle–Ottawa Scale (NOS) and the Joanna Briggs Institute (JBI) tool were used to assess methodological quality. **Results**: Six articles were included, most of them of the cohort type, with nutritional strategies lasting 2–15 weeks for overweight/obese women, based on the “macronutrient-adjusted diet” and “calorie-adjusted diet”. Only one study addressed dietary counseling in weight management, and none considered the dietary pattern. The gestational weight gain was 4.91–13.8 kg, and a lower weight gain was found in all studies that used the “macronutrient-adjusted diet” nutritional strategy. However, it did not meet the gestational weight gain targets. **Conclusions**: Despite the limited number of studies examining the impact of nutritional strategies on weight gain in women with GDM, some research suggests that diets focused on macronutrient adjustment may lead to less weight gain but are not adequate. Therefore, future studies are needed to evaluate which type of nutritional strategies ensure weight gain control during pregnancy.

## 1. Introduction

Gestational diabetes mellitus (GDM) is the most prevalent metabolic disorder during pregnancy, affecting between 3% and 25% of pregnant women worldwide [1]. Characterized primarily by hyperglycemia first detected during pregnancy, GDM can vary in severity, presenting as a transient condition or progressing to type 2 diabetes mellitus postpartum [2]. GDM is currently associated with a series of complications for both mother and child, such as fetal macrosomia, increased likelihood of cesarean delivery, and a heightened future risk of type 2 diabetes for both mother and offspring [3].

The control of maternal weight gain is recognized as an important indicator of maternal–fetal health [4], as it is associated with an increased risk of all-cause long-term mortality [5,6]. In women with GDM, the risk of such adverse outcomes may be exacerbated, also increasing large-for-gestational-age (LGA) newborns and neonatal hypoglycemia [7]. Therefore, the guidelines recommend monitoring and managing weight gain to promote maternal and fetal well-being [8].

Nutrition is one of the most influential factors in controlling weight gain in GDM cases. Diet optimization is central to treatment, contributing both to achieving glycemic control and to maintaining adequate weight gain, thus reducing the risk of complications [9]. Although there are several nutritional approaches, questions remain regarding which strategies are most effective. A previous review indicated that common dietary practices for GDM management include diets higher in complex carbohydrates and lower in fats, as well as diets enriched with extra virgin olive oil, oat bran, and supplements such as probiotics and vitamin D [10]. The benefits of these strategies have been demonstrated through other systematic reviews, which observed reduced gestational weight gain [11], lower maternal glucose levels, and decreased rates of macrosomic infants [12].

Despite promising outcomes during pregnancy, further understanding is still needed regarding prenatal nutritional strategies and their impact on weight control. Some clinical trials conducted with women with GDM have reported that dietary strategies were associated with improvements in glycemic control [13,14,15] but had no significant effect on gestational weight gain [16,17]. Clarifying this relationship is essential for developing evidence-based, effective clinical practices.

Therefore, this systematic review aims to identify prenatal nutritional strategies associated with appropriate weight gain control in women with GDM. In order to observe effects and contextual contributions and to identify risk and causality trends, this review will include observational studies. The findings of this systematic review will contribute to a more comprehensive understanding of the topic and add to the evidence previously obtained from clinical trials.

## 2. Materials and Methods

This systematic review was performed using the guidelines of the Preferred Reporting Items for Systematic Reviews and Meta-Analyses (PRISMA) [18] and registered in the International Prospective Register of Systematic Reviews (PROSPERO) under code CRD42023432322. The research question for this review was: How is diet associated with weight gain in gestational diabetes mellitus? The question was developed using the acronym PEOS (Table 1), as the review sought to understand the association between exposure and outcome without direct intervention.

### 2.1. Search Strategies and Databases

The search strategy combined terms from the Medical Subject Headings (MeSH) platform and alternative entry terms related to the population, exposure, and study types using Boolean operators. Independently, three independent reviewers (M.E.O.A., A.M.L.L., I.L.F.) carefully searched the databases. Searches were conducted in the PubMed/MEDLINE, Web of Science, Scopus, and EMBASE databases up to July 2023 without restrictions on publication date and language. A version of the strategy applied to each database is available in the Appendix A. Also, a manual search of the reference lists of selected articles was performed to identify additional relevant studies. The dataset obtained from the search was imported into the Rayyan platform to identify potential duplicate records. All records flagged as possible duplicates were manually and individually reviewed by the researchers (M.E.O.A., A.M.L.L., I.L.F.), considering the title, publication date, and abstract. Any discrepancies identified during this process were resolved by a fourth reviewer (K.D.S.R.).

### 2.2. Inclusion and Exclusion Criteria

Studies meeting the following criteria were included: (a) observational studies, including cross-sectional, cohort, and case-control designs; (b) conducted with women diagnosed with GDM; (c) provided data on diet consumed during pregnancy; and (d) reported information on gestational weight gain. Studies conducted with animals, abstracts without full text, reviews, studies involving pregnant women with type 1 or type 2 diabetes mellitus, and strategies involving physical activity were excluded.

### 2.3. Study Selection

After removing duplicate records, titles and abstracts were manually screened for eligibility by three independent reviewers (M.E.O.A., A.M.L.L., I.L.F.). This step was conducted using the Rayyan version 10.10.98 environment to facilitate the organization of record screening performed by the evaluators. The software’s suggestions regarding potential inclusion or exclusion of records were disregarded. All potentially relevant records investigating the relationship between diet in women with GDM and gestational weight gain were retrieved. For unavailable texts, the corresponding authors were contacted by email (with up to three attempts) to obtain the full text. In cases of discrepancies, a fourth reviewer (K.D.S.R.) evaluated the original publication, and following discussion, a decision was made on whether to include the record for the data extraction phase.

### 2.4. Data Extraction and Synthesis

Three independent reviewers (M.E.O.A., A.M.L.L., I.L.F.) extracted data from the included studies. If necessary, a fourth reviewer (K.D.S.R) was consulted to resolve any discrepancies during this process. The following information was extracted: author, year of publication, country, study type, number of participants, age range, pre-gestational body mass index (BMI) (kg/m^2^), nutritional strategies (food items/food groups, energy, nutrients, dietary patterns, or dietary advice), duration of follow-up for nutritional strategies, gestational age at final weight gain, outcome variables, total gestational weight gain (kg), statistical analysis, and key findings.

The nutritional strategies were categorized into two groups: “macronutrient-adjusted diets” and “calorie-adjusted diets”. Studies that examined strategies that modified the percentage of macronutrients (carbohydrates, proteins, and fats) in the prescribed diet or meal composition (main meals and snacks) were classified as “macronutrient-adjusted diets”. Meanwhile, studies focusing on caloric intake control were assigned to the “calorie-adjusted diets” group. Studies that determined total caloric intake through calculations based on the pregnant woman’s body weight—adjusted or unadjusted for pre-pregnancy BMI—or prescribed a fixed caloric intake value were included.

In studies that included pregnant women both with and without GDM, only data from the group of pregnant women with GDM were considered for the variables: number of participants, pre-gestational BMI, and gestational weight gain. Similarly, in studies that categorized pregnant women with GDM by treatment modality, only data from those treated with diet were considered. When not specified, the duration of diet monitoring was considered as the period from the patient’s entry into the study until childbirth. Gestational weight gain data were standardized in kilograms (kg).

The data from this review were presented in a narrative synthesis, describing and comparing similar findings across studies. A quantitative analysis could not be performed due to high heterogeneity among the included studies, which would compromise the validity of the meta-analysis results.

### 2.5. Assessment of Risk of Bias

The methodological quality of the studies was assessed using the Newcastle–Ottawa Scale (NOS) for cohort and case-control studies [19]. This scale consists of eight questions distributed across three domains: selection, comparability, and outcome or exposure. Each question was scored with up to 1 star, except for the comparability domain, which allowed up to 2 stars. Higher scores indicated better methodological quality, with a maximum possible score of 9 stars. The classification of methodological quality followed the criteria established by McPheeters et al. [20]. Studies scoring ≥3 stars in the “Selection” domain and ≥2 stars in the other domains were classified as “Good quality”. Studies with a score of up to 1 star in the comparability domain and up to 2 stars in the remaining domains were considered of “moderate quality”. Finally, studies scoring 0 or up to 1 star in each domain were classified as “low quality”.

For the assessment of cross-sectional studies, the Joanna Briggs Institute (JBI) tool was used [21]. This tool includes eight questions that can be answered as “Y” (Yes), “N” (No), “U” (Unclear), or “NA” (Not applicable). Final scoring classified studies as “low quality” for those with up to 3 “yes” responses, “moderate quality” for studies with 5 or 6 “yes” responses, and “high quality” for studies with 7 or 8 “yes” responses. Each study was independently evaluated by three reviewers (M.E.O.A., A.M.L.L., I.L.F.), and discrepancies were resolved by a fourth reviewer (K.D.S.R).

## 3. Results

### 3.1. Characteristics of the Studies

Initially, 724 articles were identified through database searches. Of these, 43 were selected for full-text reading, and 6 articles were ultimately included in the systematic review [22,23,24,25,26,27], as depicted in Figure 1.

### 3.2. Study Type and Population Characteristics

The majority of the included records were cohort studies (*n* = 4) [22,24,25,27] conducted between 1998 and 2014 in North America (*n* = 4) [22,23,24,25], as outlined in Table 2. The number of participants with GDM in the studies included ranged from 16 to 1129. These participants were between 29 and 35 years old, with pre-gestational BMI ranging from 22.4 to 28.8 kg/m^2^, and pre-gestational overweight was the most prevalent anthropometric nutritional status in the included studies (Table 2).

### 3.3. Nutritional Strategies Characteristics

The follow-up period for women with GDM ranged from 2 to 15 weeks (Table 3). Three studies employed the “macronutrient-adjusted diet” [22,23,24], which provided a caloric distribution of 40–45% carbohydrates, 20–25% proteins, and 30–35% lipids. Studies in this group also included modifications to meal frequency, specifying three main meals and three to four snacks per day [22,23,24].

In the “calorie-adjusted diet” group, estimated energy calculations were based on pre-pregnancy body weight and pre-gestational BMI with caloric intake ranging from 25 to 35 kcal/kg/day [25,27]. Only one study specified a fixed daily intake of 1700 kcal/day [26], while another study combined caloric adjustment of 30 kCal/kg/day with dietary counseling [27].

### 3.4. Gestational Weight Gain

#### 3.4.1. Characterization and Total Weight Gain Rate

The total weight gain reported in the included studies was based on previous information about pre-gestational weight and final weight measurements taken between 36.73 ± 5.0 and 39 ± 4 weeks of gestation. Among women with GDM, total pregnancy weight gain ranged from 4.91 ± 3.0 to 13.8 ± 6.6 kg. Studies that assessed weight gain according to pre-gestational BMI observed a gain of 4.91 ± 3.0 to 13.8 ± 6.6 kg in overweight women and 9.2 ± 3.6 to 10.0 ± 4.3 kg in eutrophic women [22,23,24].

#### 3.4.2. Macronutrient-Adjusted Diets

Morisset et al. [22], in a study with Canadian pregnant women, identified that the weekly rate of weight gain among women with GDM was significantly lower (−0.17 kg/week, *p* ≤ 0.0001) in the third trimester compared to pregnant women without GDM. Additionally, total weight gain in the third trimester was lower among women with GDM (−2.24 kg, *p* ≤ 0.0001) when compared to the control group, comprising women without GDM. Morisset et al. [23] conducted a second study in Canada; in this instance, the study employed a case-control design. Consistent with the latest observations, the authors observed that the weekly weight gain rate in women with GDM was significantly lower (0.30 ± 0.27 vs. 0.61 ± 0.50 kg/week, *p* ≤ 0.05) during the third trimester compared to women with normal glucose tolerance.

Similarly, Couch et al. [24], in a study conducted in the United States, also demonstrated that pregnant women with GDM had lower weight gain across all gestational weeks compared to women with a negative diabetes screening test during pregnancy (12.17 ± 4.98 vs. 15.19 ± 4.87, *p* < 0.0001).

#### 3.4.3. Calorie-Adjusted Diets

In the study by Most and Langer [25], women with GDM consuming calorie-controlled diets gained less weight compared to those treated with insulin (10.2 ± 7.7 vs. 11.1 ± 8.6, *p* = 0.003). However, the results of Sunjaya et al. [26], in a cross-sectional study conducted in Indonesia, did not identify a significant difference in weight gain between the groups of women treated with a 1700 kcal diet, insulin, and oral antidiabetics (4.91 ± 3.0 vs. 5.10 ± 5.7 and 1.93 ± 3.3; *p* = 0.867). Similarly, Ho et al. [27] also found no significant difference (*p* > 0.05) when evaluating pregnant women according to tertiles of caloric intake (Tertile 1: 9.2 ± 3.6; Tertile 2: 10.0 ± 3.8; Tertile 3: 10.0 ± 4.3; *p* > 0.05).

### 3.5. Risk of Bias of the Studies

The average score of the cohort and case-control studies on the NOS was rated as moderate, with a mean of 6 stars. The lower scores were attributed to the ‘selection’ domain in cohort studies and the ‘comparability’ domain in case-control studies. The methodological quality assessment of the cross-sectional study, based on the JBI criteria, indicated a moderate quality level, with a total of 5 ‘yes’ responses (Appendix A).

## 4. Discussion

### 4.1. Main Findings

Our systematic review gathered data from six studies [22,23,24,25,26,27] conducted with women with GDM who were followed with nutritional strategies prescribed during prenatal care and had their weight gain evaluated. In three studies [22,23,24], the strategies were based on the distribution of macronutrients and meal distribution throughout the day, and there was less weight gain among pregnant women under this dietary recommendation when compared with other treatments or situations.

The most prevalent nutritional status observed in the analyzed studies was pre-gestational overweight. The reported total weight gain ranged widely, from 4.91 ± 3.0 to 13.8 ± 6.6 kg, and was associated with pre-gestational BMI. Overweight women, for example, experienced greater weight gain (4.91 ± 3.0 to 13.8 ± 6.6 kg) compared to eutrophic women (9.2 ± 3.6 to 10.0 ± 4.3 kg) [22,23,24,25,26,27]. Conversely, pregnant women with GDM, particularly those adhering to macronutrient-adjusted diets, exhibited lower weight gain during the third trimester [22,23,24]. These findings underscore the complexity of controlling gestational weight gain and highlight the need to tailor dietary strategies to the profiles of pregnant women. Therefore, a comprehensive dietary approach is needed to improve the management of GDM and prevent complications associated with inadequate weight gain.

### 4.2. Comparison with Existing Literature

A previous systematic review, which included randomized clinical trials conducted with women with GDM, demonstrated that a method based on the energy distribution of macronutrients in meals contributed to reducing gestational complications, including excessive weight gain [31]. These findings are supported by the studies included in this review [22,23,24,25,26,27], which highlighted the positive impact of macronutrient-adjusted diets on the management of gestational weight gain. Therefore, adjustments in energy distribution and macronutrient intake can be effective strategies to help pregnant women achieve recommended weight goals, thereby minimizing risks associated with excessive weight gain.

A retrospective cohort study involving 382 women diagnosed with GDM investigated the effects of dietary treatment based on calorie restriction, following the Danish national guidelines for diabetes. Participants were offered a diet of 1434 kcal/day, 50% of which consisted of carbohydrates. Total gestational weight gain ranged from 8.7 ± 7 to 14.0 ± 6 kg, and the findings revealed that women who required insulin initiation gained more weight compared to those managed solely with the dietary strategy [32]. Among the studies included in this review, Sunjaya et al. [26] reported no significant association between weight gain and treatment groups, including calorie-adjusted diets, insulin therapy, and oral antidiabetic medications. Conversely, Most and Langer [25], who based their dietary approach on calorie calculations according to pre-pregnancy BMI, achieved a lower total weight gain compared to interventions involving insulin therapy. These findings highlight the inconsistency in gestational weight outcomes when the focus of the nutritional strategy is on total dietary caloric control.

Although most of the studies included [22,23,24,25,26,27] achieved lower gestational weight gain among women with GDM who were treated with nutritional strategies, the recommendation range established by the IOM [28] for overweight pregnant women was still not reached in three of the studies [22,23,24]. Since dietary guidelines are primarily based on macronutrient composition and meal distribution throughout the day, the data also highlight the need for diets that focus on dietary patterns, emphasizing the qualitative aspects of the diet.

According to the World Health Organization (WHO), a healthy eating pattern can be understood as the sum of foods based on availability, accessibility, preferences, culture, traditions, and other factors. It is essential to incorporate the fundamental principles of food, such as adequacy, diversity, balance, and moderation [33,34]. Additionally, the need for variety within food groups, as well as the foods that make up these groups, is emphasized, as this plays a significant role in promoting health through dietary diversity. Regarding carbohydrate intake, it should primarily come from whole grains, vegetables, and minimally processed fruits, with recommendations to consume two servings of fruits and vegetables and 25 g of fiber per day. Concerning lipid intake, saturated and trans fatty acids should be reduced and replaced by unsaturated fatty acids, particularly polyunsaturated ones, which should be prioritized [33,34]. Moreover, it is important to note that the American Diabetes Association [8], for the management of GDM, recommends lifestyle changes, including physical activity, as well as the use of insulin when glycemic targets are not achieved.

### 4.3. Strengths and Limitations

To the best of our knowledge, this is the first systematic review of observational studies that evaluates the relationship between prenatal nutritional strategies and gestational weight gain in women with GDM, providing a compilation of information on dietary interventions currently being implemented in clinical practice based on observational data. The nutritional strategies discussed in this review may provide guidance for healthcare professionals, particularly dietitians, in the treatment of women with GDM. Implementing these strategies alongside actions focused on dietary and nutritional education may assist in enhancing patients’ understanding of their nutrition within this context. Furthermore, continuous follow-up by a multidisciplinary team may be essential to achieving glycemic control goals and offering support to patients, thereby minimizing potential challenges of treatment during pregnancy.

The studies included [22,23,24,25,26,27] evaluated weight gain in women with GDM who were treated with diet alone, separating them from groups without GDM or those treated with other modalities, such as insulin and oral antidiabetics. Additionally, none of the studies mentioned the monitoring of physical activity. Therefore, it was possible to assess the isolated relationship between nutritional strategies and gestational weight gain, independent of other treatments.

The limitations of this review include the low representativeness of the study samples and the short duration of the dietary interventions, with one study lasting only 2 weeks, which may be considered insufficient for evaluating the dietary impact. Another important aspect is the limited variation in the types of nutritional strategies evaluated, most of which are based on macronutrients and meal distribution throughout the day. For example, the influence of dietary patterns or the consumption of other nutrients was not addressed, highlighting the need for studies that evaluate this perspective in this population.

Additionally, some references used for the dietary guidelines were not identified, making it unclear how the dietary counseling was implemented in the study that employed this intervention. Furthermore, two studies lacked information on the food survey methods used, preventing an assessment of adherence to the diet. Finally, a statistical analysis would be useful to determine whether participants who received nutritional strategies are less likely to experience inadequate weight gain compared to those who did not achieve satisfactory adherence to dietary treatment, as the studies evaluated only presented comparative measures.

## 5. Conclusions

Although data in the literature are limited and the quality of the studies is moderate, our systematic review suggests that a nutritional strategy focused on macronutrient adjustment is associated with reduced weight gain in women with GDM compared to those without GDM or those undergoing other treatment modalities. However, it is important to note that gestational weight gain still fails to meet the parameters established for pregnant women. These findings highlight the need for further prospective cohort studies in populations with diverse dietary contexts to enable a meta-analysis of the data and better understand the effects of nutritional strategies on weight gain. Additionally, studies are required to compare the effects of nutritional strategies with other treatments, assess diet quality, and evaluate adherence to the strategy and its effectiveness in managing weight gain. This approach will provide robust evidence to guide nutritional treatment in GDM and achieve positive maternal–infant outcomes.

## Figures and Tables

**Figure 1 nutrients-17-00043-f001:**
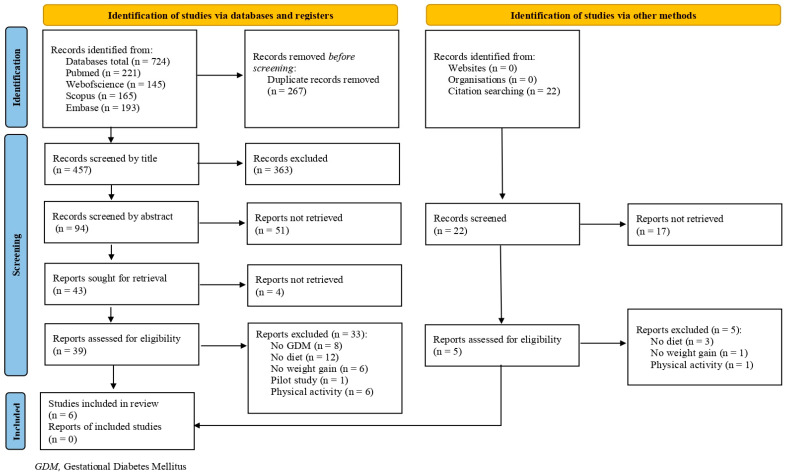
A flow diagram of included studies.

**Table 1 nutrients-17-00043-t001:** Search strategy used in the systematic review.

Items	Description
Population (P)	Women diagnosed with gestational diabetes mellitus
Exposure (E)	Nutritional strategies applied to maternal diet
Outcome (O)	Gestational weight gain
Study types (S)	Observational studies

**Table 2 nutrients-17-00043-t002:** Characteristics of the studies and population included in the systematic review on the relation of nutritional strategies on weight gain in GDM.

Authors (Year)	Country	Study Type	*nº* of Participants	Pre-pregnancy BMI (kg/m^2^)	Nutritional Strategies	Gestational Age of Final Weight Gain (Weeks)	Total Gestational Weight Gain (kg)	Outcome Variable	Effect of Nutritional Strategies on Gestational Weight Gain	Main Findings of the Studies
Morisset et al. [22] (2011)	Canada	Cohort	55	28.2 ± 7.6	Macronutrient-adjusted diet	37	~13	Total gestational weight gain per quarter and per week in kg	Lower weight gain in women with GDM during the third trimester compared to women without GDM (*p* ≤ 0.0001).	Weight gain in women with GDM was greater in the first trimester (*p* ≤ 0.01) and less in the third trimester when compared to the control group (*p* ≤ 0.0001).
Morisset et al. [23] (2014)	Canada	Control- case	17	28.8 ± 8.0	Macronutrient-adjusted diet	38.6 ± 1.3	13.8 ± 6.6	Total gestational weight gain (kg)	Lower weight gain in women with GDM during the third trimester compared to women without GDM (*p* ≤ 0.05).	Women with GDM who adhered to the proposed nutritional strategy had significantly lower energy (*p* = 0.05) and carbohydrate (*p* = 0.001) intake compared to women with normal glucose tolerance.
Couch et al. [24] (1998)	USA	Cohort	25	25.6 ± 6.3	Macronutrient-adjusted diet	37–38	12.17 ± 4.98	Total gestational weight gain (kg)	Lower total weight gain in women with GDM compared to women with a negative diabetes screening result during pregnancy (*p* < 0.0001).	Women with GDM exhibited higher levels in biochemical parameters (HbA1c, FFAs, TG, VLDL, and HDL-TG associated with cholesterol) and hormonal markers (plasma progesterone, prolactin, and β-estradiol) compared to the control group (*p* < 0.05).
Most and Langer [25] (2012)	USA	Cohort	1129	Healthy weight, 26%; overweight, 35%; obesity, 39%	Calorie-adjusted diets	39 ± 4	10.2 ± 7.7	Total gestational weight gain (kg)	Lower total weight gain in women with GDM compared to women treated with insulin (*p* = 0.003).	Overweight pregnant women who followed nutritional strategies and maintained glycemic control were four times more likely to have LGA newborns (OR = 4.08, IC de 95% 2.8–5.9; *p* < 0.05).
Sunjaya and Sunjaya [26] (2019)	Indonesia	Cross-sectional	16	27.76 ± 3.83	Calorie-adjusted diets	36.73 (±5.0)	4.91 ± 3.0	Average weight gain (kg)	No difference in weight gain was observed between groups (*p* = 0.867).	Women with GDM using oral antidiabetic medications had higher fasting blood glucose levels compared to those managed with nutritional strategies (*p* = 0.045) or insulin therapy (*p* = 0.045).
Ho et al. [27] (2005)	China	Cohort	62	22.4 ± 3.2–23.1 ± 4.2	Calorie-adjusted diets + dietary counseling	Over 38	9.2 ± 3.6–10.0 ± 4.3	Total gestational weight gain (kg)	No difference in weight gain was observed between groups (*p* > 0.05)	No differences were observed in glycemic profile characteristics or neonatal outcomes across tertiles of caloric intake.

Note. USA = United States of America; BMI = body mass index; GDM = gestational diabetes mellitus; LGA = large-for-gestational-age.

**Table 3 nutrients-17-00043-t003:** Characteristics of the Nutritional Strategies included in the systematic review.

Authors (Year)	Description of the Nutritional Strategy	Measurement of the Nutritional Strategy	Duration of Dietary Follow-Up (Weeks)	Methods of Food Consumption Assessment	Nutritional Strategy References
Morisset et al. [22] (2011)	Carbohydrates (40% to 45%), proteins (20% to 25%), lipids (30% to 35%), and meal distribution (3 main meals and 3 snacks)	kCal/day for energy and g/day for macronutrients	~15	Three 24 h recalls	NR
Morisset et al. [23] (2014)	Carbohydrates (40% to 45%), proteins (20% to 25%), lipids (30% to 35%), and meal distribution (3 main meals and 3 snacks)	kCal/day for energy and g/day for macronutrients	14	Food frequency questionnaire (FFQ): 91 items and 33 sub-items	Dietary Reference Intakes of Institute of Medicine [28]
Couch et al. [24] (1998)	Carbohydrates (45%), proteins (22%), lipids (33%), and meal distribution (3 main meals and 3 snacks)	kCal/day for energy and g/day for macronutrients	10–15	24 h recall	Standard clinical protocol at Hartford Hospital [29]
Most and Langer [25] (2012)	Diet of 25 kCal/kg (overweight and obesity) or 35 kCal/kg (healthy weight), carbohydrates (40%), and meal distribution (3 main meals and 4 snacks per day)	NR	2	NR	Clinical management guidelines for obstetrician–gynecologists [30]
Sunjaya and Sunjaya [26] (2019)	1700 kCal diet	NR	NR	NR	NR
Ho et al. [27] (2005)	30 kCal/kg diet + dietary counseling	Caloric intake tertile	10	Weighted food diary—5 days	NR

Note. NR = not reported.

## Data Availability

The data that support the findings of the review are derived from publicly available sources. All included studies and datasets were obtained from published research articles, which are accessible via databases. No new primary data were generated in the course of this research.

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
