# Peer review of "Nutritional Strategies Prescribed During Pregnancy and Weight Gain in Women with Gestational Diabetes Mellitus: A Systematic Review of Observational Studies"

_nutrients, 2024, doi:10.3390/nu17010043_

Round 1

Reviewer 1 Report

Comments and Suggestions for Authors It s a good systematic review on an important but not very highlited subject. The choice and quality of articles shows the lack of good original studies on this topic. Unfortunately the conclusions are not very novel and important but it sometimes like that. I would emphasize to write more about your own strategies in the Discussion to make it little more interesting and educative

Author Response

  1. Comments 1: “It s a good systematic review on an important but not very highlited subject. The choice and quality of articles shows the lack of good original studies on this topic. Unfortunately the conclusions are not very novel and important but it sometimes like that. I would emphasize to write more about your own strategies in the Discussion to make it little more interesting and educative”

Response 1: We appreciate your valuable comment. We reflected on your assessment and decided to include a paragraph emphasizing the importance of the nutritional strategies observed in the review, particularly regarding their impact on the clinical practice of healthcare professionals. We also highlighted the importance of multidisciplinary follow-up. The change can be found on lines 312-318.

Reviewer 2 Report

Comments and Suggestions for Authors

In their systematic review, the Authors aimed to identify diets related to weight gain in pregnant women diagnosed with gGDM. According to the Preferred Reporting Items for Systematic Reviews and Meta-analyses, and searching in common database (PubMed/MEDLINE, Web of Science, Scopus and EMBASE), they identified six articles. in only one  study addressed dietary counseling in weight management, and none considered the dietary pattern. A gestational weight gain of 4.91 - 13.8 kg and a lower weight gain were found in all studies that used the “Macronutrient-adjusted diet” nutritional strategy. so, the Authors concluded that, despite the limited number of studies examining the impact of nutritional strategies on weight gain in women with GDM, diets focused on macronutrient adjustment may lead to less weight gain but not adequate, requiring future studies to evaluate which type of nutritional strategies ensures weight gain control during pregnancy.

The manuscript has a good research question, beyond being original. It is well-written, with appropriate references, and correct methodological approach. Results have been expressed correctly. Discussion is well-performed, including strengths and limitations. In the opinion of this reviewer, it could be stimulating for future prospective studies.

Author Response

  1. Comment 1: In their systematic review, the Authors aimed to identify diets related to weight gain in pregnant women diagnosed with gGDM. According to the Preferred Reporting Items for Systematic Reviews and Meta-analyses, and searching in common database (PubMed/MEDLINE, Web of Science, Scopus and EMBASE), they identified six articles. in only one study addressed dietary counseling in weight management, and none considered the dietary pattern. A gestational weight gain of 4.91 - 13.8 kg and a lower weight gain were found in all studies that used the “Macronutrient-adjusted diet” nutritional strategy. so, the Authors concluded that, despite the limited number of studies examining the impact of nutritional strategies on weight gain in women with GDM, diets focused on macronutrient adjustment may lead to less weight gain but not adequate, requiring future studies to evaluate which type of nutritional strategies ensures weight gain control during pregnancy.

The manuscript has a good research question, beyond being original. It is well-written, with appropriate references, and correct methodological approach. Results have been expressed correctly. Discussion is well-performed, including strengths and limitations. In the opinion of this reviewer, it could be stimulating for future prospective studies.

 Response 1: Thank you for your comment and assessment. We appreciate your recognition of our dedication in conducting this review.

Reviewer 3 Report

Comments and Suggestions for Authors

Ln 40: "...as a transient or persistent condition postpartum" - Rephrase, otherwise confusing: gestational vs postpartum....???

Ln 37 to 52: may be more brief

Ln 93: Rayyan software - Details please - also this is AI software

Ln 102-103: "three ...reviewers...of the Rayyan software" = Unclear

Conclusion = Weak

Author Response

  1. Comment 1: Ln 40: "...as a transient or persistent condition postpartum" - Rephrase, otherwise confusing: gestational vs postpartum....???

 Response 1: We appreciate the opportunity for review. We noticed that the wording of this sentence was unclear and have revised line 40 to: “Characterized primarily by hyperglycemia first detected during pregnancy, GDM can vary in severity, presenting as a transient or progressing to type 2 diabetes mellitus postpar-tum

  1. Comment 2: Ln 37 to 52: may be more brief

Response 2: We appreciate your observation. We have chosen to keep the first paragraph unchanged, as it succinctly addresses relevant aspects of GDM, such as definition, epidemiology, and adverse repercussions. However, we realized that the second paragraph could be improved, and thus, we made revisions, which can be found in lines 44 to 49.

  1. Comment 3: Ln 93: Rayyan software - Details please - also this is AI software

 Response 3: We appreciate the opportunity to review the text. During this review, we used the Rayyan platform solely for the duplicate removal process, which was done manually and individually, without influencing the reviewers' decision-making. The changes explaining the use of Rayyan are in lines 89-94.

  1. Comment 4: Ln 102-103: "three ...reviewers...of the Rayyan software" = Unclear

Response 4: We appreciate your observation. The software's indications for including and excluding records, as well as identifying the most relevant records, were not considered during the screening and selection stages. These steps were done manually and individually, without influencing the reviewers' decision-making. The changes explaining the use of Rayyan are in lines 104-108.

  1. Comment 5: Conclusion = Weak

Response 5: We appreciate the opportunity to review the text. We have rewritten the last paragraph to strengthen our findings. We also emphasize the importance of conducting more cohort studies, performing meta-analyses, and exploring the qualitative aspect of maternal diet (lines 347-353).